# The Impact of Race–Ethnicity and Diagnosis of Alzheimer’s Disease and Related Dementias on Mammography Use

**DOI:** 10.3390/cancers14194726

**Published:** 2022-09-28

**Authors:** Aokun Chen, Yongqiu Li, Jennifer N. Woodard, Jessica Y. Islam, Shuang Yang, Thomas J. George, Elizabeth A. Shenkman, Jiang Bian, Yi Guo

**Affiliations:** 1Department of Health Outcomes and Biomedical Informatics, College of Medicine, University of Florida, Gainesville, FL 32611, USA; 2Moffit Cancer Center, Tampa, FL 33612, USA

**Keywords:** screening, mammography, electronic health record, cognitive impairment, social determinant of health

## Abstract

**Simple Summary:**

Analyzing real-world data from the OneFlorida+ Clinical Research Network, we examined the impact of ADRD diagnosis and race–ethnicity on mammography use in breast cancer screening (BCS)-eligible women. We found that BCS-eligible women with Alzheimer’s disease (AD) and AD-related dementias (ADRD) were more likely to undergo a mammography than the BCS-eligible women without ADRD. Stratified by race–ethnicity, BCS-eligible Hispanic women with ADRD were more likely to undergo a mammography, whereas BCS-eligible non-Hispanic black and non-Hispanic other women with ADRD were less likely to undergo a mammography.

**Abstract:**

Breast cancer screening (BCS) with mammography is a crucial method for improving cancer survival. In this study, we examined the association of Alzheimer’s disease (AD) and AD-related dementias (ADRD) diagnosis and race–ethnicity with mammography use in BCS-eligible women. In the real-world data from the OneFlorida+ Clinical Research Network, we extracted a cohort of 21,715 BCS-eligible women with ADRD and a matching comparison cohort of 65,145 BCS-eligible women without ADRD. In multivariable regression analysis, BCS-eligible women with ADRD were more likely to undergo a mammography than the BCS-eligible women without ADRD (odds ratio [OR] = 1.19, 95% confidence interval [CI] = 1.13–1.26). Stratified by race–ethnicity, BCS-eligible Hispanic women with ADRD were more likely to undergo a mammography (OR = 1.56, 95% CI = 1.39–1.75), whereas BCS-eligible non-Hispanic black (OR = 0.72, 95% CI = 0.62–0.83) and non-Hispanic other (OR = 0.65, 95% CI = 0.45–0.93) women with ADRD were less likely to undergo a mammography. This study was the first to report the impact of ADRD diagnosis and race–ethnicity on mammography use in BCS-eligible women using real-world data. Our results suggest ADRD patients might be undergoing BCS without detailed guidelines to maximize benefits and avoid harms.

## 1. Introduction

Breast cancer is the most prevalent cancer and the second leading cause of death by cancer in women in the United States (US) [1]. It is estimated that approximately 287,850 women will be diagnosed with and 43,250 women will die from breast cancer in the United States in 2022 [1]. Breast cancer screening (BCS) with mammography is a crucial method for improving cancer survival. Undergoing mammography helps detect breast cancer in its early stages, and thus provides the best opportunity for successful treatment and prognosis. The benefits of screening mammography have been well-documented in many randomized trials and observation studies [1,2,3,4,5,6,7,8,9]. In the report from the International Agency for Research on Cancer (IARC), mammography reduced the mortality rate of breast cancer by 40% among women of 50–69 years of age [9]. The proven benefits of mammography led to the creation of BCS guidelines by national professional associations, such as the U.S. Preventive Services Task Force (USPSTF) and the American Cancer Society [5,10,11,12,13]. For instance, the USPSTF recommends that women aged 50 to 74 years get a screening mammography every two years [5].

Even with the BCS guidelines, the decision to undergo a mammography is not always straightforward, especially for patients with Alzheimer’s disease (AD) and AD-related dementias (ADRD). The existing guidelines do not provide tailored recommendations for patients with specific conditions, including ADRD [5]. Currently, no official guidelines are provided for ADRD patients, let alone tailored for the ADRD patients at different stages. On the other hand, ADRD patients have a shortened life expectancy (<10 years), which reduces the cumulative benefits of BCS [14,15,16,17,18,19]. Further, living with behavioral and psychiatric disorders, poor functional status, and comorbidities, ADRD patients are more likely to experience complications from cancer screening and the follow-up procedures [20,21]. As a result, the use of mammography could be very different among ADRD patients as compared to the general population. To date, there exists only one study on the association between ADRD and mammography use, but the association was examined in female breast cancer patients. In that study, Weng et al. analyzed the Surveillance, Epidemiology, and End Results (SEER)–Medicare data and found that breast cancer patients with ADRD were less likely to receive BCS than those without ADRD in a retrospective design [22]. There is a need for more studies, especially prospective studies, on the association between ADRD diagnosis and mammography use in BCS-eligible populations.

In this study, we examined the impact of ADRD diagnosis on mammography use among BCS-eligible women using real-world data, including those in electronic health records (EHRs) and claims, from the OneFlorida+ Clinical Research Network (CRN) [23], which is a large CRN in the national PCORnet. We extracted a cohort of BCS-eligible women with ADRD in OneFlorida+ and prospectively examined whether mammography was used following ADRD diagnosis. In addition, we examined whether there existed any racial–ethnic disparities in the association of ADRD diagnosis with mammography use. EHRs are real-time, patient-centered medical records that have gained popularity in medical research. Compared with datasets collected under restricted eligibility rules (e.g., SEER–Medicare), EHRs reflect the real-world clinical outcome of ADRD patients, making it easier to translate and apply the computational methods and research findings into practice. To the best of our knowledge, this was the first study to report the impact of ADRD diagnosis and race–ethnicity on mammography use in BCS-eligible women using real-world data.

## 2. Materials and Methods

### 2.1. Data Source and Study Population

This study was approved by the University of Florida Institutional Review Board. We obtained 2012–2021 EHR data from the OneFlorida+ CRN, one of the eight CRNs in the national PCORnet funded by the Patient-Centered Outcomes Research Institute (PCORI). Our study population included a case cohort of BCS-eligible women with ADRD and a matching comparison cohort of BCS-eligible women without ADRD. To create the case cohort, we extracted all women with an ADRD diagnosis between the ages of 50 and 72 years from the OneFlorida+ EHRs. The International Classification of Diseases (ICD) codes used to identify ADRD, including AD, vascular dementia, Lewy body dementia, or frontotemporal dementia, were listed in Appendix A Table A1. We considered an upper age limit of 72 years, rather than 74 years as recommended by the USPSTF, because we included a two-year observation window in the outcome definition (see Section 2.2). For the case cohort, the index date was defined as the date of the first ADRD diagnosis. We excluded case patients who had no encounters one year before and two years after the index date to allow the observation of the baseline measures and BCS outcome. We further excluded case patients who were previously diagnosed with breast cancer prior to the index date, as well as patients without a valid ZIP code. To create the comparison cohort, we extracted all women without any ADRD diagnosis, dementia (e.g., alcohol-induced persisting dementia, drug-induced persisting dementia), or conditions that could cause dementia (e.g., Parkinson’s disease, paralysis agitans, temporal sclerosis, etc.; see Appendix A Table A1 for ICD codes) between the ages of 50 and 73 years from the OneFlorida+ EHRs. To these women, we applied the same exclusion criteria that were applied to the case cohort. Next, each case woman with ADRD was matched to three women without ADRD based on age and calendar year. For the comparison cohort, the index date was defined as the date of a random inpatient or outpatient encounter in the matched calendar year.

### 2.2. Study Outcome and Exposures

The primary outcome of the study was mammography use (yes or no) within two years (i.e., 730 days) after the index date (Figure 1). Since the USPSTF recommended that women aged 50 to 74 years should get a screening mammography every two years, we considered a woman in our study population compliant with the USPSTF guideline if she underwent a mammography within two years after the index date. Mammography use within the outcome observation window was identified using Current Procedural Terminology (CPT) codes 77065, 77066, and 77067. The primary exposures were having an ADRD diagnosis or not (i.e., belonging to the case or comparison cohort) and race–ethnicity (non-Hispanic white [NHW], non-Hispanic black [NHB], non-Hispanic other [NHO], Hispanic, or unknown).

### 2.3. Covariates

We included in our data analysis a number of covariates that could potentially impact the utilization of mammography. These covariates included age, comorbidity, rurality (i.e., urban vs. rural residency), and social vulnerability. Comorbidity was measured using the Charlson Comorbidity Index (CCI), with a higher CCI indicating worse baseline health [24]. To calculate the CCI, we extracted the following conditions at baseline from the EHRs (Figure 1): moderate/severe liver disease, cerebrovascular disease, peripheral vascular disease, renal disease, hemiplegia or paraplegia, dementia, mild liver diseases, congestive heart failure, chronic obstructive pulmonary disease, diabetes, and diabetes with complications. The ICD codes used to identify these conditions were listed in Appendix A Table A2. We included rurality in the study as previous studies showed disparities in medical resources accessibility between rural and urban patients that may affect their cancer screen use [25,26]. Rurality was measured using the ZIP code level rural–urban commuting area (RUCA) codes. RUCA codes are a census tract-based classification that characterize all U.S. census tracts with respect to their rural/urban status and commuting relationships to other census tracts [27]. ZIP code level RUCA codes are approximations of that at the census tract level, in which each ZIP code is assigned with a primary code. The primary codes range between 1 and 10, with 1 being the most urbanized and 10 being rural. We linked the RUCA codes to our study population using the patients’ latest ZIP codes in the EHRs. Social vulnerability was measured using the Center for Disease Control and Prevention (CDC)’s Social Vulnerability Index (SVI). The SVI uses U.S. census data to measure the social vulnerability of every census tract. Each census tract is ranked on 15 social factors and grouped into four related themes that include socioeconomic status (SVI-SS), household composition and disability (SVI-HCD), minority status and language (SVI-MSL), and housing type and transportation (SVI-HTT). A higher value of SVI indicates higher social vulnerability. We linked the SVI as the four SVI themes to our study population using the patients’ latest ZIP code in EHRs.

### 2.4. Statistical Analysis

First, to characterize our study population, we calculated the means with standard deviations (SDs) and frequencies with percentages for the variables of interest stratified by ADRD diagnosis. Second, to examine the association between ADRD diagnosis and mammography use, we built a logistic regression model (base model) with mammography use being the dependent variable and ADRD diagnosis, race–ethnicity, and the covariates being the independent variables. Third, to examine whether race–ethnicity modified the association between ADRD diagnosis and mammography use, we extended the base model to include an additional ADRD diagnosis by race–ethnicity interaction term (interaction model). Results from both logistic models were reported as odds ratios (ORs) and the associated 95% confidence intervals (CIs). All statistical analyses were conducted using python 3.9.4 and SAS 9.4 (SAS Institute Inc., Cary, NC, USA).

## 3. Results

### 3.1. Characteristics of Study Population

We summarized the characteristics of our study population in Table 1. Overall, we extracted 21,715 BCS-eligible women with ADRD and 65,145 matching BCS-eligible women without ADRD from OneFlorida+. The ADRD and non-ADRD patients had the same mean age (64.8 vs. 64.8 years, *p* = 1.000), the matching variable. Although the overall racial–ethnic makeup did not differ significantly between the two study cohorts (*p* = 0.114), a higher percentage of NHW (47.1% vs. 42.7%) and NHO (4.9% vs. 3.5%) patients were presented in the non-ADRD comparison cohort compared with the ADRD patient cohort. Compared to the non-ADRD patients, the ADRD patients were more likely to reside in urban areas (91.3% vs. 87.8%, *p* < 0.001), have worse baseline health (CCI > 0: 28.5% vs. 15.8%, *p* < 0.001), and associate with higher social vulnerability in terms of all four SVI themes (SVI-SS: 0.57 vs. 0.55, *p* < 0.001; SVI-HCD: 0.52 vs. 0.49, *p* < 0.001; SVI-MSL: 0.60 vs. 0.59, *p* < 0.001; SVI-HTT: 0.57 vs. 0.50, *p* < 0.001). Lastly, the BCS-eligible ADRD patients were more likely to undergo a mammography within two years of the index date than the BCS-eligible non-ADRD patients (10.0% vs. 8.0%, *p* < 0.001).

We summarized the social vulnerability of the study population by race in Table 2. Among BCS-eligible women with ADRD, in SVI-SS, SVI-HCD, SVI-HTT, NHW (SVI-SS 0.61 [0.20]; SVI-HCD: 0.55 [0.19]; SVI-HTT 0.60 [0.15]) were the more socially vulnerable compared to NHW (SVI-SS 0.52 [0.19]; SVI-HCD: 0.54 [0.19]; SVI-HTT 0.54 [0.15]), NHO (SVI-SS 0.51 [0.20]; SVI-HCD: 0.50 [0.17]; SVI-HTT 0.52 [0.17]), and Hispanic (SVI-SS 0.59 [0.20]; SVI-HCD: 0.47 [0.18]; SVI-HTT 0.58 [0.18]). Hispanic BCS-eligible women with ADRD were more socially vulnerable in SVI-MSL (0.77 [0.22]) compared to NHW (0.46 [0.23]), NHB (0.60 [0.23]), and NHO (0.55 [0.24]). The same trend could be found among BCS-eligible women without ADRD (NHW: SVI-SS: 0.47 [0.20], SVI-HCD: 0.54 [0.19], SVI-MSL: 0.46 [0.23], SVI-HTT: 0.54 [0.16]; NHB: SVI-SS: 0.61 [0.20], SVI-HCD: 0.55 [0.19], SVI-MSL: 0.60 [0.23], SVI-HTT: 0.60 [0.15]; NHO: SVI-SS: 0.51 [0.20], SVI-HCD: 0.50 [0.17], SVI-MSL: 0.55 [0.24], SVI-HTT: 0.52 [0.17]; Hispanic: SVI-SS: 0.59 [0.20], SVI-HCD: 0.47 [0.18], SVI-MSL: 0.77 [0.22], SVI-HTT: 0.58 [0.18]). NHW, NHO, and Hispanic BCS-eligible women with ADRD were more socially vulnerable compared to those without ADRD, while NHB BCS-eligible women without ADRD were more socially vulnerable than those who had ADRD.

### 3.2. Multivariable Logistic Regression Analysis

We summarized the ORs and the associated 95% CIs from the base and interaction models in Table 2. In the base model, we observed that the BCS-eligible women with ADRD were more likely to undergo a mammography within two years after the index date than the BCS-eligible women without ADRD (OR = 1.19, 95% CI = 1.13–1.26), adjusting for the covariates. Compared with the NHW patients, the NHB (OR = 1.43, 95% CI = 1.33–1.55), NHO (OR = 1.24, 95% CI = 1.08–1.41), and Hispanic (OR = 1.32, 95% CI = 1.22–1.42) patients were more likely to undergo a mammography. Regarding the covariates, ADRD patients who were older (OR = 0.95, 95% CI = 0.95–0.96) and had worse baseline health (CCI > 1 vs. = 0: OR = 0.81, 95% CI = 0.72–0.90; CCI = 1 vs. = 0: OR = 0.85, 95% CI = 0.79–0.92) were less likely to undergo a mammography. SVI-SS was positively associated with mammography use (OR = 0.77, 95% CI = 0.60–0.99), while SVI-MSL (OR = 1.96, 95% CI = 1.68–2.27) and SVI-HTT (OR = 1.59, 95% CI = 1.30–1.94) were negatively associated with mammography use. We did not observe any urban–rural differences in mammography use.

In the interaction model, we found a significant ADRD diagnosis by the race–ethnicity interaction term and thus reported the OR for ADRD diagnosis in each of the racial–ethnic groups (Table 3). Mammography use did not differ by ADRD diagnosis in BCS-eligible NHW women. However, among Hispanic women who were BCS-eligible, those with ADRD were more likely to undergo a mammography within two years after the index date than those without ADRD (OR = 1.56, 95% CI = 1.39–1.75), adjusting for the covariates. In contrast, among NHB women who were BCS-eligible, those with ADRD were less likely to undergo a mammography than those without ADRD (OR = 0.72, 95% CI = 0.62–0.83). A similar finding was observed among NHO women who were BCS-eligible (OR = 0.65, 95% CI = 0.45–0.93). Regarding the covariates, similar ORs as those reported in the base model were obtained in the interaction model. ADRD patients who were older (OR = 0.95, 95% CI = 0.94–0.96) and had worse baseline health (CCI > 1 vs. = 0: OR = 0.83, 95% CI = 0.74–0.93; CCI = 1 vs. = 0: OR = 0.86, 95% CI = 0.79–0.93) were less likely to undergo a mammography. SVI-SS was positively associated with mammography use (OR = 0.77, 95% CI = 0.60–0.99), while minority status and language (OR = 1.89, 95% CI = 1.62–2.19) and housing type and transportation (OR = 1.62, 95% CI = 1.32–1.99) were negatively associated with mammography use. We did not observe any urban–rural differences in mammography use.

## 4. Discussion

In this study, we extracted a cohort of BCS-eligible women with ADRD and a matching comparison cohort of BCS-eligible women without ADRD in the OneFlorida+ CRN. We examined the association of ADRD diagnosis and race–ethnicity with mammography use adjusting for age, baseline health, rurality, and social vulnerability. We found that BCS-eligible women with ADRD were more likely to undergo a mammography than the BCS-eligible women without ADRD. The BCS rate among the control group was low compared to that of the general BCS eligible population. This difference was expected, as the control group was selected to match the age, encounter year, and month of the AD patients. The matching criteria made the BCS-use pattern among the control group different from that in the general population. Stratified by race–ethnicity, BCS-eligible Hispanic women with ADRD were more likely to undergo a mammography, whereas BCS-eligible non-Hispanic black and non-Hispanic other women with ADRD were less likely to undergo a mammography. Mammography use did not differ by ADRD diagnosis in BCS-eligible NHW women.

Unexpectedly, we found that BCS-eligible women with ADRD were more like to use mammography than those without ADRD. Although no studies have examined mammography use in BCS-eligible ADRD patients, previous research has shown that mammography use is lowered in BCS-eligible, cognitively impaired women [28]. Our finding of an overall positive association between ADRD diagnosis and mammography use appears to be inconsistent with research on the cognitively impaired. However, our interaction analysis revealed that this observation was driven by a similar positive association between ADRD diagnosis and mammography use in BCS-eligible Hispanic women only. Lower or same likelihood of mammography use was observed in NHW, NHB, or NHO women with ADRD. As undergoing BCS for ADRD patients is usually decided through a shared decision-making process involving the physicians, patients, and their caregivers, more future studies are needed to confirm the observed racial–ethnic and ADRD disparities in mammography use and explore the psychosocial mechanisms behind these disparities. Considering ADRD patients have a shortened life expectancy and reduced quality of life, they were less likely to benefit from BCS. Our study provided the first evidence that ADRD patients might be undergoing BCS without detailed guidelines to maximize benefits and avoid harms

As a clarification, our study does not discourage the use of BCS among ADRD patients. We suggest further studies to improve the benefit of BCS for ADRD patients in different stages. On the one hand, ADRD patients were shown to have good health baselines, i.e., low CCI according to our study cohort, qualifying them as a beneficial group of BCS. On the other, the estimated survival of diagnosed ADRD patients was around 3 to 8 years [29,30], potentially making them less beneficial for BCS.

Our study has a few strengths. A major strength is the use of real-world data in a large CRN in the national PCORnet. The rich data in OneFlorida+ allowed us to accurately identify our study population of ADRD and non-ADRD patients, their detailed medical history, and mammography use. Further, we used a prospective study design that examined mammography use within two years after ADRD diagnosis. Our study also has a few limitations to note. Due to the observational nature of the study, our results do not support any causal relationship between ADRD diagnosis and mammography use. In addition, we were unable to control for some potentially important covariates such as education or income due to the limitations of the EHR data. However, the SVI contains components that cover many of these variables at the fine-grained ZIP code level. Moreover, our results may not reflect the BCS use for all ADRD patients. As ADRD is usually diagnosed at later stages, our results may not reflect BCS use among early staged ADRD patients.

## 5. Conclusions

In summary, this study was the first to report the impact of ADRD diagnosis and race–ethnicity on mammography use in BCS-eligible women using real-world data. Our finding that BCS-eligible women, especially Hispanic women, with ADRD were more like to use mammography suggests that ADRD patients might be getting screened for BCS despite a lack of clear benefits. More future studies are needed to explore the psychosocial mechanisms behind these disparities in mammography use.

## Figures and Tables

**Figure 1 cancers-14-04726-f001:**
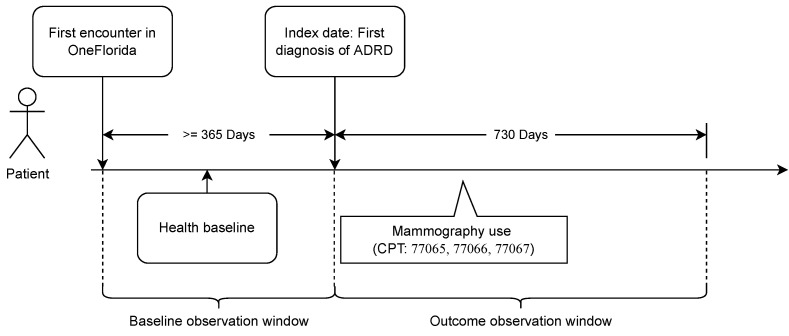
Index date and observation window for the study population.

**Table 1 cancers-14-04726-t001:** Characteristics of the study population.

	BCS-Eligible Women with ADRD n = 21,715	BCS-Eligible Women without ADRD n = 65,145	*p*-Value
**Age**			
Mean (SD)	64.8 (6.4)	64.8 (6.4)	=1.000
**Race–ethnicity**			=0.114
NHW	9261 (42.7%)	30,700(47.1%)	
NHB	3545 (16.3%)	9919 (15.2%)	
NHO	758 (3.5%)	3198 (4.9%)	
Hispanic	4365 (20.1%)	8809 (20.7%)	
Unknown	3786 (17.4%)	8156 (12.5%)	
**Rurality**			<0.001
Urban	19,829 (91.3%)	57,266 (87.9%)	
Rural	1886 (8.7%)	7879 (12.1%)	
**Charlson Comorbidity Index**			<0.001
0	15,544 (71.6%)	54,914(82.3%)	
1	3987 (18.4%)	6759 (10.4%)	
>1	2184 (10.1%)	3472 (5.3%)	
**Social Vulnerability Index**			
SVI-SS (SD)	0.57 (0.20)	0.55 (0.21)	<0.001
SVI-HCD (SD)	0.52 (0.19)	0.49 (0.19)	<0.001
SVI-MSL (SD)	0.60 (0.26)	0.59 (0.26)	<0.001
SVI-HTT (SD)	0.57 (0.17)	0.50 (0.17)	<0.001
**Mammography use**			<0.001
Yes	2074 (10.0%)	4999 (8.0%)	
No	19,641 (90.0%)	60,146 (92.0%)	

BCS: breast cancer screening; ADRD: Alzheimer’s disease and related dementia; SD: standard deviation; NHW: non-Hispanic white; NHB: non-Hispanic black; NHO: non-Hispanic other; SVI: social vulnerability index; SS: socioeconomic status; HCD: household composition and disability; MSL: minority status and language; HTT: housing type and transportation.

**Table 2 cancers-14-04726-t002:** Social vulnerability of the study population by race.

	BCS-Eligible Women with ADRD n = 21,715	BCS-Eligible Women without ADRD n = 65,145
**NHW**		
SVI-SS (SD)	0.52 (0.19)	0.47 (0.20)
SVI-HCD (SD)	0.54 (0.19)	0.49 (0.20
SVI-MSL (SD)	0.46 (0.23)	0.48 (0.23)
SVI-HTT (SD)	0.54 (0.16)	0.49 (0.16)
**NHB**		
SVI-SS (SD)	0.61 (0.20)	0.63 (0.20)
SVI-HCD (SD)	0.55 (0.19)	0.56 (0.18)
SVI-MSL (SD)	0.60 (0.23)	0.61 (0.23)
SVI-HTT (SD)	0.60 (0.15)	0.59 (0.16)
**NHO**		
SVI-SS (SD)	0.51 (0.20)	0.47 (0.20)
SVI-HCD (SD)	0.50 (0.17)	0.46 (0.19)
SVI-MSL (SD)	0.55 (0.24)	0.57 (0.23)
SVI-HTT (SD)	0.52 (0.17)	0.48 (0.16)
**Hispanic**		
SVI-SS (SD)	0.59 (0.20)	0.57 (0.20)
SVI-HCD (SD)	0.47 (0.18)	0.46 (0.18)
SVI-MSL (SD)	0.77 (0.22)	0.75 (0.21)
SVI-HTT (SD)	0.58 (0.18)	0.55 (0.18)
**Unknown**		
SVI-SS (SD)	0.60 (0.20)	0.59 (0.20)
SVI-HCD (SD)	0.48 (0.17)	0.50 (0.18)
SVI-MSL (SD)	0.75 (0.24)	0.69 (0.27)
SVI-HTT (SD)	0.59 (0.17)	0.58 (0.17)

NHW: non-Hispanic white; NHB: non-Hispanic black; NHO: non-Hispanic other; SVI: social vulnerability index; SVI: social vulnerability index; SS: socioeconomic status; HCD: household composition and disability; MSL: minority status and language; HTT: housing type and transportation.

**Table 3 cancers-14-04726-t003:** Multivariable logistic regression estimating the association between ADRD diagnosis and mammography use in women aged 50–73 in OneFlorida+.

Variables	Base Model		Interaction Model	
	Adjusted OR (95% CI)	*p*-Value	Adjusted OR (95% CI)	*p*-Value
* **Primary Exposures** *				
**ADRD diagnosis**				
Yes vs. No	1.19 (1.13, 1.26)	<0.001	-	-
**Race–ethnicity**				
NHB vs. NHW	1.43 (1.33, 1.55)	<0.001	-	-
Hispanic vs. NHW	1.32 (1.22, 1.42)	<0.001	-	-
NHO vs. NHW	1.24 (1.08, 1.41)	=0.016	-	-
Unknown vs. NHW	3.06 (2.85, 3.29)	<0.001	-	-
**ADRD diagnosis by** **Race–ethnicity**				
Yes vs. No in NHW	-	-	1.04 (0.94, 1.15)	=0.479
Yes vs. No in NHB	-	-	0.72 (0.62, 0.83)	<0.001
Yes vs. No in NHO	-	-	0.65 (0.45, 0.93)	=0.019
Yes vs. No in Hispanics	-	-	1.56 (1.39, 1.75)	<0.001
Yes vs. No in Unknown	-	-	1.55 (1.40, 1.71)	<0.001
* **Covariates** *				
**Age**	0.95 (0.95, 0.96)	<0.001	0.95 (0.95, 0.96)	<0.001
**Charlson Comorbidity Index**				
>1 vs. 0	0.81 (0.72, 0.90)	=0.001	0.83 (0.74, 0.93)	=0.009
1 vs. 0	0.85 (0.79, 0.92)	=0.001	0.86 (0.79, 0.93)	=0.002
**Rurality**				
Urban vs. Rural	0.96 (0.88, 1.06)	=0.449	0.98 (0.89, 1.08)	=0.651
**Social Vulnerability Index**				
SVI-SS	0.77 (0.60, 0.99)	=0.038	0.77 (0.60, 0.99)	=0.040
SVI-HCD	0.98 (0.80, 1.20)	=0.832	0.99 (0.80, 1.21)	=0.900
SVI-MSL	1.96 (1.68, 2.27)	<0.001	1.89 (1.62, 2.19)	<0.001
SVI-HTT	1.59 (1.30, 1.94)	<0.001	1.62 (1.32, 1.99)	<0.001

OR: odds ratio; CI: confidence interval; ADRD: Alzheimer’s disease and related dementias; NHW: non-Hispanic white; NHB: non-Hispanic black; NHO: non-Hispanic other; SVI: social vulnerability index; SS: socioeconomic status; HCD: household composition and disability; MSL: minority status and language; HTT: housing type and transportation.

## Data Availability

The OneFlorida+ electronic health record data are considered Protected Health Information under the Health Insurance Portability and Accountability Act of 1996 (HIPAA) in the United States, and as such are not publicly available.

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
