# Peer review of "The Impact of Race–Ethnicity and Diagnosis of Alzheimer’s Disease and Related Dementias on Mammography Use"

_cancers, 2022, doi:10.3390/cancers14194726_

Round 1

Reviewer 1 Report

Please see file.

Author Response

Review 1

This is a very innovative and potentially useful article. We know little about mammography screening in the face of morbidities that may be limiting, such as ADRD. The choice of samples, including construction of the comparison sample, and methods seem sound.

There are limitations to the data available that may have implications for the authors conclusions, however. Specifically, this involves the use of the CDC’s Social Vulnerability Index (SVI). While it is reasonable to use data at the census tract level, we are told nothing about how the SVI, especially the SVI-SS, varies by race within the two samples. This could be crucial in interpreting results. The authors conclude/assume that screening mammography is of less benefit in women with ADRD. I will address this in the next paragraph, but they find that BCS-eligible Hispanic women with ADRD are more likely to get the screening within the first two years after ADRD diagnosis than NHB and NHO women in the samples. What is missing is information on how SVI varies between racial groups and especially whether there are insurance differences between these groups. This may very well be the case. A diagnosis of ADRD likely qualifies women without health insurance for Medicaid insurance and perhaps other healthcare benefits. It may be the case that Hispanic women are more likely to receive insurance benefits when they did not have them previously after the ADRD diagnosis. More likely, however, is that Hispanic family income ($82,917 per year) is much higher in Florida than NH Black family income ($63,773). Hispanic is much closer to white (NHW) family income ($89,469). I have done this only cursorily, but it needs to be explored and included. The authors conclude that Hispanic women and NHW women with ADRD are receiving breast cancer screening with no clear benefit. I would argue that women with ADRD and in fact all women with cognitive impairment have as much right to breast cancer diagnosis and treatment as women without these impairments. So, while I agree that it may be the case that these women are being exploited to increase providers’ incomes, I disagree with the notion that there is not benefit to being diagnosed and treated.

Reply to the reviewer

Thanks for raise these suggestions.

For the SVI variation among racial groups, we found that NHO, NHB, and Hispanic patients had worse health baseline than that of NHW. We have added the table and description below in our revision.

SVI

BCS-eligible women with ADRD
n = 21,715

BCS-eligible women without ADRD
n = 65,145

NHW

SVI-SS (SD)

0.52 (0.19)

0.47 (0.20)

SVI-HCD (SD)

0.54 (0.19)

0.49 (0.20

SVI-MSL (SD)

0.46 (0.23)

0.48 (0.23)

SVI-HTT (SD)

0.54 (0.16)

0.49 (0.16)

NHB

SVI-SS (SD)

0.61 (0.20)

 0.63 (0.20)

SVI-HCD (SD)

0.55 (0.19)

0.56 (0.18)

SVI-MSL (SD)

0.60 (0.23)

0.61 (0.23)

SVI-HTT (SD)

0.60 (0.15)

0.59 (0.16)

NHO

SVI-SS (SD)

0.51 (0.20)

0.47 (0.20)

SVI-HCD (SD)

0.50 (0.17)

0.46 (0.19)

SVI-MSL (SD)

0.55 (0.24)

0.57 (0.23)

SVI-HTT (SD)

0.52 (0.17)

0.48 (0.16)

Hispanic

SVI-SS (SD)

0.59 (0.20)

0.57 (0.20)

SVI-HCD (SD)

0.47 (0.18)

0.46 (0.18)

SVI-MSL (SD)

0.77 (0.22)

0.75 (0.21)

SVI-HTT (SD)

0.58 (0.18)

0.55 (0.18)

Unknown

SVI-SS (SD)

0.60 (0.20)

0.59 (0.20)

SVI-HCD (SD)

0.48 (0.17)

0.50 (0.18)

SVI-MSL (SD)

0.75 (0.24)

0.69 (0.27)

SVI-HTT (SD)

0.59 (0.17)

0.58 (0.17)

We summarized the social vulnerability of the study population by race in Table 2. Among BCS-eligible women with ADRD, in SVI-SS, SVI-HCD, SVI-HTT, NHW (SVI-SS 0.61 [0.20]; SVI-HCD: 0.55 [0.19]; SVI-HTT 0.60 [0.15]) were the more socially vulnerable compared to NHW (SVI-SS 0.52 [0.19]; SVI-HCD: 0.54 [0.19]; SVI-HTT 0.54 [0.15]), NHO (SVI-SS 0.51 [0.20]; SVI-HCD: 0.50 [0.17]; SVI-HTT 0.52 [0.17]), and Hispanic (SVI-SS 0.59 [0.20]; SVI-HCD: 0.47 [0.18]; SVI-HTT 0.58 [0.18]). Hispanic BCS-eligible women with ADRD were more socially vulnerable in SVI-MSL (0.77 [0.22]) compared to NHW (0.46 [0.23]), NHB (0.60 [0.23]), and NHO (0.55 [0.24]). The same trend could be found among BCS-eligible women without ADRD (NHW: SVI-SS: 0.47 [0.20], SVI-HCD: 0.54 [0.19], SVI-MSL: 0.46 [0.23], SVI-HTT: 0.54 [0.16]; NHB: SVI-SS: 0.61 [0.20], SVI-HCD: 0.55 [0.19], SVI-MSL: 0.60 [0.23], SVI-HTT: 0.60 [0.15]; NHO: SVI-SS: 0.51 [0.20], SVI-HCD: 0.50 [0.17], SVI-MSL: 0.55[0.24], SVI-HTT: 0.52 [0.17]; Hispanic: SVI-SS: 0.59 [0.20], SVI-HCD: 0.47 [0.18], SVI-MSL: 0.77 [0.22], SVI-HTT: 0.58 [0.18]). NHW, NHO, and Hispanic BCS-eligible women with ADRD were more socially vulnerable compared to those without ADRD, while NHB BCS-eligible women without ADRD were more socially vulnerable than those who had ADRD.

For the insurance variation among racial groups, we did not find huge difference among NHB, Hispanic, and NHW ADRD patients. However, due to the high missing rate, we cannot further conduct statistic analysis on the insurance and BCS use.

Insurance

BCS-eligible women with ADRD
n = 21,715

BCS-eligible women without ADRD
n = 65,145

NHW

Medicare/Medicaid

8716 (88.3%)

15753 (51.3%)

Private

550 (5.9%)

5197 (16.9%)

Other Government

45 (0.5%)

425 (1.4%)

Unknown

490 (5.3%)

9325 (30.4%)

NHB

Medicare/Medicaid

3250 (91.7%)

6475 (65.3%)

Private

136 (3.8%)

1437 (14.5%)

Other Government

7 (0.2%)

95 (1.0%)

Unknown

152 (4.3%)

1912 (19.2%%)

NHO

Medicare/Medicaid

644 (85.0%)

1725 (53.9%)

Private

47 (6.2%)

516 (16.1%)

Other Government

7 (0.9%)

31 (1.0%)

Unknown

60 (7.9%)

926 (29.0%)

Hispanic

Medicare/Medicaid

3961 (90.7%)

8545 (64.9%)

Private

199 (4.6%)

1767 (13.4%)

Other Government

15 (0.3%)

148 (1.1%)

Unknown

190 (4.4%)

2712 (20.6%)

Unknown

Medicare/Medicaid

3643 (96.2%)

7479 (91.7%)

Private

16 (3.8%)

177 (2.2%)

Other Government

2 (0.1%)

9 (0.1%)

Unknown

125 (3.3%)

491 (6.0%)

For the BCS benefits in ADRD patients, we further clarified our point with the following description in the discussion:

To clarify, our study does not discourage the use of BCS among ADRD patients, but suggest for further studies to improve the benefit of BCS among ADRD patients at different stages. On one hand, ADRD patients were shown to have a better health baseline, i.e., low CCI according to our study cohort, qualifying them as a beneficial group of BCS. On the other, the estimated survival of diagnosed ADRD patients was around 3 to 8 years1,2, making them less beneficial for BCS.

Smaller concerns include that:
• A citation is needed for the statement that women with ADRD are more likely to experience
complications from cancer screening and the follow-up procedures (page 2, lines 60-63).

Reply to the reviewer

Thanks for pointing out the issue. We will add the following literature as citations:

Advani S, Braithwaite D. Optimizing selection of candidates for lung cancer screening: role of comorbidity, frailty and life expectancy. Transl Lung Cancer Res. 2019 Dec;8(Suppl 4):S454-S459. doi: 10.21037/tlcr.2019.10.03. PMID: 32038937; PMCID: PMC6987350.

Zhang Y, Bian J, Huo J, Yang S, Guo Y, Shao H. Comparing the downstream costs and healthcare utilization associated with the use of low-dose computed tomography (LDCT) in lung cancer screening in patients with and without alzheimer's disease and related dementias (ADRD). Curr Med Res Opin. 2021 Oct;37(10):1731-1737. doi: 10.1080/03007995.2021.1953972. Epub 2021 Jul 26. PMID: 34252317; PMCID: PMC8627644.

  • It would be good to include examples (or more wording) on that lack of recommendations
    included with BCS guidelines for women with “specific conditions, including ADRD” (page 2,
    lines58, 59).

Reply to the reviewer

Thanks for the suggestion. We have added the following example to the paragraph:

Currently, no official guidelines are provided for ADRD patients let along tailored for the ADRD patients at different stages.

  • In line 74 on page 2, the authors say they used “real-world data, including those in electronic
    health records (EHR), claims, and more........” What do they mean by “more”?

Reply to the reviewer:

Thanks for pointing out this error. We have changed our description to:

real-world data, including those in electronic health records (EHR) and claims

  • I am confused about why they expected differences between rural and urban women. More
    narrative is needed here. What differences were they expecting?

Reply to the reviewer:

Thanks for your advice, we have added the following description to the paper:

We included rurality into the study as the previous study showed disparity in medical resources accessibility between rural and urban patients that may affect their cancer screen use.

We cited the following literatures to support this:

  1. Douthit, N.; Kiv, S.; Dwolatzky, T.; Biswas, S. Exposing Some Important Barriers to Health Care Access in the Rural USA. Public Health 2015, 129, 611–620, doi:10.1016/j.puhe.2015.04.001.
  2. Levit, L.A.; Byatt, L.; Lyss, A.P.; Paskett, E.D.; Levit, K.; Kirkwood, K.; Schenkel, C.; Schilsky, R.L. Closing the Rural Cancer Care Gap: Three Institutional Approaches. JCO Oncology Practice 2020, 16, 422–430, doi:10.1200/OP.20.00174.

Reviewer 2 Report

The Impact of Race-Ethnicity and Diagnosis of Alzheimer’s Disease and Related Dementias on Mammography Use

Simple Summary: Recommend that you spell out Alzheimer’s Disease and Related Dementias before using the abbreviation ADRD.

Abstract: Can it really be said that there no evidence of benefit of mammography screening in women with ADRD? The issue is never any particular disease, but rather the ability to undergo mammography, and expected longevity.

Page 2, line 49: My advice is that the authors cite the RCT evidence without the meta-analysis point estimate because it does not reflect the mortality reductions observed with modern mammography. The RCTs demonstrated the efficacy of screening for breast cancer with an intention to treat analysis, but they are a poor estimate of the effectiveness of screening among women who attend screening.  It is OK to cite the Nelson meta-analysis, but the summary RR is a poor example of evidence-based medicine when the trials vary so much in the degree to which they achieved a reduction in the incidence rate of advanced disease. 1  Better to cite the recent IARC report (which is available as a free download) for estimates of the benefit of screening among women who attend screening.2

Line 59: Is 10 years an average estimate of longevity from the first indication of Alzheimer's? That is a long time to judge early detection and minimal therapy would not be worthwhile. The question arises, if a woman can successfully complete mammography, wouldn’t it be better to detect breast cancer early than at an advanced state several years later, when she still may have 10+ years of life remaining, undergoing significant quality of life diminishing therapy? I’m not suggesting that this isn’t very complicated, only that this early in the argument, dropping overall average longevity, vs. “depending on the age and stage of Alzheimer’s may have very limited longevity,”  seems premature. One other point. Looking at Table 1, the majority of these women have CCI’s of 0. Is it possible to bring in the issue of longevity among Alzheimer’s patients in the context of CCI?

Line 99: What comes to mind is whether the index data of the first diagnosis of Alzheimer’s is a good proxy of the “onset” of Alzheimer's. Is there any way to adjust for the degree of Alzheimer's, since it seems some cases can be quite advanced by the time a diagnosis is delivered.

Line 178: This study population is 50-73, and only 8% of the control group had a mammogram in the past 2 years? That is surprisingly low, and worthy of some explanation.

Line 179: Would the authors describe the insurance coverage for this population? Is there any chance that it might affect uptake or referral to screening?

Line 248: The 10 year average longevity seems arbitrary. There is a range of age of onsets, and a range of duration of Alzheimer’s, and the progression, severity, and disabilities vary. I think the authors are positioning the discussion to raise the question “should Alzheimer's patients be screened for breast cancer at all because, on average, they have limited longevity?” What the author’s have not shown is how many patients may have been severely disabled by their disease and had very limited longevity due to ADRD, or ADRD + other chronic conditions. Is the argument that screening is acceptable until there is only an estimated 10 years left to live?  I’m reminded that the study evaluates mammography use in the 2 years after the diagnosis of ADRD…..isn’t it possible that the symptoms, on average, may not have been severe during that period, compared with afterwards, when screening rates may have been lower? Then again, there is the fact that screening is surprisingly very low in both groups, which needs to be addressed in the findings.   

1.            Tabar L, Yen AM, Wu WY, et al. Insights from the breast cancer screening trials: how screening affects the natural history of breast cancer and implications for evaluating service screening programs. Breast J. Jan-Feb 2015;21(1):13-20. doi:10.1111/tbj.12354

2.            IARC Working Group on the Evaluation of Cancer-Preventive Strategies. Breast cancer screening, Volume 15. vol 15. IARC Handbooks of Cancer Prevention. IARC Press; 2016.

Author Response

Review 2

The Impact of Race-Ethnicity and Diagnosis of Alzheimer’s Disease and Related Dementias on Mammography Use

Simple Summary: Recommend that you spell out Alzheimer’s Disease and Related Dementias before using the abbreviation ADRD.

Reply to the reviewer:

Thanks for pointing out this issue. We have replaced ADRD with Alzheimer's Disease (AD) and AD-related dementias (ADRD)

Abstract: Can it really be said that there no evidence of benefit of mammography screening in women with ADRD? The issue is never any particular disease, but rather the ability to undergo mammography, and expected longevity.

Reply to the reviewer

Thanks for raising this issue. We have rephrased the current description “Our results suggest ADRD patients might be getting screened for BCS despite a lack of clear benefits.” To “Our results suggest ADRD patients might be getting screened for BCS despite a lack of detailed guidelines to maximize benefits.”

Page 2, line 49: My advice is that the authors cite the RCT evidence without the meta-analysis point estimate because it does not reflect the mortality reductions observed with modern mammography. The RCTs demonstrated the efficacy of screening for breast cancer with an intention to treat analysis, but they are a poor estimate of the effectiveness of screening among women who attend screening.  It is OK to cite the Nelson meta-analysis, but the summary RR is a poor example of evidence-based medicine when the trials vary so much in the degree to which they achieved a reduction in the incidence rate of advanced disease. 1  Better to cite the recent IARC report (which is available as a free download) for estimates of the benefit of screening among women who attend screening.2

Reply to the reviewer

Thanks for the advice and material. We have added the IARC report and replace the description from “In the meta-analysis conducted by Nelson et al, mammography reduced the mortality rate of breast cancer by 19% to 21% among women 40 – 75 years of age” to “In the report from International Agency for Research on Cancer (IARC), mammography reduced the mortality rate of breast cancer by 35% among women 50 – 69 years of age.”

Line 59: Is 10 years an average estimate of longevity from the first indication of Alzheimer's? That is a long time to judge early detection and minimal therapy would not be worthwhile. The question arises, if a woman can successfully complete mammography, wouldn’t it be better to detect breast cancer early than at an advanced state several years later, when she still may have 10+ years of life remaining, undergoing significant quality of life diminishing therapy? I’m not suggesting that this isn’t very complicated, only that this early in the argument, dropping overall average longevity, vs. “depending on the age and stage of Alzheimer’s may have very limited longevity,”  seems premature. One other point. Looking at Table 1, the majority of these women have CCI’s of 0. Is it possible to bring in the issue of longevity among Alzheimer’s patients in the context of CCI?

Reply to the reviewer

Thanks for raising this issue. We have added this point into the discussion with the following writings:

As a clarification, our study does not discourage the use of BCS among ADRD patients, but suggest for further studies to improve the benefit of BCS among ADRD patients at different stages. On one hand, ADRD patients were shown to have a better health baseline, i.e., low CCI according to our study cohort, qualifying them as a beneficial group of BCS. On the other, the estimated survival of diagnosed ADRD patients was around 3 to 8 years1,2, making them less beneficial for BCS.

We will cite the following literature for ADRD patient survival:

  1. Mueller C, Soysal P, Rongve A, Isik AT, Thompson T, Maggi S, Smith L, Basso C, Stewart R, Ballard C, O'Brien JT, Aarsland D, Stubbs B, Veronese N. Survival time and differences between dementia with Lewy bodies and Alzheimer's disease following diagnosis: A meta-analysis of longitudinal studies. Ageing Res Rev. 2019 Mar;50:72-80. doi: 10.1016/j.arr.2019.01.005. Epub 2019 Jan 6. PMID: 30625375.
  2. Brookmeyer R, Corrada MM, Curriero FC, Kawas C. Survival Following a Diagnosis of Alzheimer Disease. Arch Neurol. 2002;59(11):1764–1767. doi:10.1001/archneur.59.11.1764

Line 99: What comes to mind is whether the index date of the first diagnosis of Alzheimer’s is a good proxy of the “onset” of Alzheimer's. Is there any way to adjust for the degree of Alzheimer's, since it seems some cases can be quite advanced by the time a diagnosis is delivered.

Reply to the reviewer

Thanks for point out this issue. We will include this as a limitation of the study with the description below.

Also, our results may not reflect the BCS use for all ADRD patients. As ADRD are usually diagnosed at later stages, our results may not reflect BCS use among early staged ADRD patients.

Line 178: This study population is 50-73, and only 8% of the control group had a mammogram in the past 2 years? That is surprisingly low, and worthy of some explanation.

Reply to the reviewer

Thanks for pointing out this point. We have added the following description to the paper in result section:

The BCS rate among the control group was low compared to that of the general BCS eligible population. This difference was expected as the control group was selected to match the age, encounter year and month of an AD patient. The matching criteria made the BCS use pattern among the control group different from that in the general population.

Line 179: Would the authors describe the insurance coverage for this population? Is there any chance that it might affect uptake or referral to screening?

Reply to the reviewer

Thanks for raise this point. However, we are not able to assess the insurance coverage for this population. In our study cohort, we observed that 18.8% of all the patients did not have a valid insurance information. The missing in insurance made it hard for us to make solid conclusion on how insurance affect BCS use.

Line 248: The 10 year average longevity seems arbitrary. There is a range of age of onsets, and a range of duration of Alzheimer’s, and the progression, severity, and disabilities vary. I think the authors are positioning the discussion to raise the question “should Alzheimer's patients be screened for breast cancer at all because, on average, they have limited longevity?” What the author’s have not shown is how many patients may have been severely disabled by their disease and had very limited longevity due to ADRD, or ADRD + other chronic conditions. Is the argument that screening is acceptable until there is only an estimated 10 years left to live?  I’m reminded that the study evaluates mammography use in the 2 years after the diagnosis of ADRD…..isn’t it possible that the symptoms, on average, may not have been severe during that period, compared with afterwards, when screening rates may have been lower? Then again, there is the fact that screening is surprisingly very low in both groups, which needs to be addressed in the findings.  

Reply to the reviewer

Thanks for pointed out this point. We have addressed the issues in our reply to line 59 and line 178

  1. Tabar L, Yen AM, Wu WY, et al. Insights from the breast cancer screening trials: how screening affects the natural history of breast cancer and implications for evaluating service screening programs. Breast J. Jan-Feb 2015;21(1):13-20. doi:10.1111/tbj.12354
  2. IARC Working Group on the Evaluation of Cancer-Preventive Strategies. Breast cancer screening, Volume 15. vol 15. IARC Handbooks of Cancer Prevention. IARC Press; 2016.

Round 2

Reviewer 2 Report

cancers-1821129

The authors modified a statement to say “Our results suggest ADRD patients might be getting screened for BCS despite a lack of detailed guidelines to maximize benefits.”

Just to be clearer, I would suggest saying,  “Our results suggest ADRD patients are undergoing BCS without detailed guidelines to maximize benefits and avoid harms.”  Overall guidance is needed, but it seems it is important to be clear whether an Alzheimer’s patient is in good health and early stages of Alzheimer’s vs. more advanced stage Alzheimer’s.  Despite the distinction, the authors are correct that there is no guidance.

Regarding the reduction of breast cancer mortality, the authors should be clear that they are referring to incidence-based mortality cohort studies of programmatic screening have shown the among women who attend screening, breast cancer mortality is approximately 40% lower compared with women who do not attend screening. (not 35%; page 457 in the IARC Handbook).

Line 285: suggest rather than saying “making them less beneficial for BCS,” saying “potentially making BCS less beneficial for them.”  I think everyone will understand “potentially,” as meaning, in all instances, careful consideration should be given to preventive health measures that may not be beneficial in the long term, and may be harmful in the short term.

Author Response

Reply to the Cancers reviewer

Thanks for your valuable opinion and suggestion for the paper. We have included our reply in red and highlighted our changes in the manuscript.

The authors modified a statement to say “Our results suggest ADRD patients might be getting screened for BCS despite a lack of detailed guidelines to maximize benefits.”

Just to be clearer, I would suggest saying,  “Our results suggest ADRD patients are undergoing BCS without detailed guidelines to maximize benefits and avoid harms.”  Overall guidance is needed, but it seems it is important to be clear whether an Alzheimer’s patient is in good health and early stages of Alzheimer’s vs. more advanced stage Alzheimer’s.  Despite the distinction, the authors are correct that there is no guidance.

Reply:

Thanks for the suggestion. We have revised our manuscript.

Regarding the reduction of breast cancer mortality, the authors should be clear that they are referring to incidence-based mortality cohort studies of programmatic screening have shown the among women who attend screening, breast cancer mortality is approximately 40% lower compared with women who do not attend screening. (not 35%; page 457 in the IARC Handbook).

Reply:

Thanks for point out this error. We have corrected the breast cancer morality reduction number.

Line 285: suggest rather than saying “making them less beneficial for BCS,” saying “potentially making BCS less beneficial for them.”  I think everyone will understand “potentially,” as meaning, in all instances, careful consideration should be given to preventive health measures that may not be beneficial in the long term, and may be harmful in the short term.

Reply:

Thanks for this suggestion, we have added “potentially” into our manuscript.